# Synthesis and Pharmacological Evaluation of Novel 2,3,4,5-tetrahydro[1,3]diazepino[1,2-*a*]benzimidazole Derivatives as Promising Anxiolytic and Analgesic Agents

**DOI:** 10.3390/molecules26196049

**Published:** 2021-10-06

**Authors:** Dmitriy V. Maltsev, Alexander A. Spasov, Pavel M. Vassiliev, Maria O. Skripka, Mikhail V. Miroshnikov, Andrey N. Kochetkov, Nataliya V. Eliseeva, Yuliya V. Lifanova, Tatyana A. Kuzmenko, Lyudmila N. Divaeva, Anatolii S. Morkovnik

**Affiliations:** 1Department of Pharmacology and Bioinformatics, Volgograd State Medical University, 400131 Volgograd, Russia; aspasov@mail.ru (A.A.S.); pvassiliev@mail.ru (P.M.V.); rete.mirabile.renis@gmail.com (M.O.S.); mihailmiroshnkov@rambler.ru (M.V.M.); nvkirillova@rambler.ru (N.V.E.); imjerry@mail.ru (Y.V.L.); 2Laboratory of Experimental Pharmacology, Volgograd Medical Scientific Center, 400131 Volgograd, Russia; 3Laboratory for Information Technology in Pharmacology and Computer Modeling of Drugs for Reasearch Center of Innovative Medicines, Volgograd Medical Scientific Center, 400131 Volgograd, Russia; akocha@mail.ru; 4Research Institute of Physical and Organic Chemistry, Laboratory of organic synthesis, Southern Federal University, 344090 Rostov-on-Don, Russia; takuzmenko@sfedu.ru (T.A.K.); divaevaln@mail.ru (L.N.D.); asmork2@ipoc.sfedu.ru (A.S.M.)

**Keywords:** diazepino[1,2-*a*]benzimidazole, triazabenzo[*a*]cyclopenta[*cd*]azulenium, anxiolytic, analgesic, hot plate, open field, tail flick, elevated plus maze, docking analysis

## Abstract

A number of novel 2,3,4,5-tetrahydro[1,3]diazepino[1,2-*a*]benzimidazole derivatives **2** were obtained by alkylation mainly in the 1*H*-tautomeric form of 2,3,4,5-tetrahydro[1,3]diazepino[1,2-*a*]benzimidazole or its 8,9-dimethyl-substituted analog 4-chlorobenzyl bromide, 4-chloroacetic acid fluoroanilide, and 4-tert-butylphenacyl bromide in neutral medium. Compounds 3 were cyclized and synthesized earlier with 11-phenacyl-substituted diazepino[1,2-*a*]benzimidazoles upon heating in conc. HBr. The chemical structures of the compounds were clarified by using the ^1^H Nuclear Magnetic Resonance Spectroscopy (^1^H-NMR) technique. Anxiolytic properties were evaluated using the elevated plus maze (EPM) and open field (OF) tests. The analgesic effect of compounds was estimated with the tail flick (TF) and hot plate (HP) methods. Besides, possible the influence of the test compounds on motor activities of the animals was examined by the Grid, Wire, and Rotarod tests. Compounds **2d** and **3b** were the most active due to their prominent analgesic and anxiolytic potentials, respectively. The results of the performed in silico analysis showed that the high anxiolytic activity of compound **3b** is explained by the combination of a pronounced interaction mainly with the benzodiazepine site of the GABA_A_ receptor with a prominent interaction with both the specific and allosteric sites of the 5-HT_2A_ receptor.

## 1. Introduction

At present, the recognition of the unity of biological, psychological, and social components of some pathologies, such as anxiety disorders [1] and pain syndrome [2], in addition to their multifactorial etiology, has led to a significant increase in interest among researchers globally. It is well known that anxiety and fear are evolutionary mechanisms of self-defense and survival, in addition to the nociceptive system, which has an adaptive–limiting function [3]. Therefore, a low level of anxiety and pain sensitivity cannot only be perceived as an exclusively positive phenomenon, but it is also necessary to consider the degree of danger of such a state [4]. However, one should distinguish the background level of anxiety from pathological anxiety, the symptoms of which include irritability, tension and inability to relax, a feeling of “nervousness”, memory impairment, and sleep disturbance [5]. The somatic manifestations of anxiety are highly diverse and include chronic pain of muscle hypertension, cardiovascular, and even pseudo-allergic symptoms [6]. According to Gal et al. [7], there is feedback between chronic pain syndrome and the formation of anxiety, depression, and sleep disturbance: it is necessary to combine analgesics and anxiolytics to correct these conditions [8]. Despite the adaptive function of pain, the chronic nature of acute pain is not adaptive and leads to the development of clinical syndromes [9]. In turn, complaints of acute and chronic pain remain difficult to treat, and there is a need for new effective analgesics in clinical practice [2,10]. This is a consequence of the complexity and diversity of pain modulation mechanisms at the pathophysiological level [2], in which many different mediators and receptors are involved, in addition to the heterogeneity of pain syndromes and the unreliability of some preclinical pain models [10]. Furthermore, the different etiology of the subtypes of anxiety disorders is due to the low efficiency of the use of existing anxiolytics in a number of clinical cases [11].

In the course of a directed search for active substances among the chemical class of imidazobenzimidazole derivatives, a compound under the code RU-31 was identified, possessing 5-HT_2A_-antagonistic and anti-anxiety properties [12,13]. The behavioral activity of animals under the influence of compound RU-1276 with 5-HT_3_-antagonistic action [14], with a single administration in the “Elevated Plus Maze” test, was characteristic of the action of anxiolytics (an increase both in the number of exits into the open arms of the EPM and the total time spent in them). In addition, compound RU-476, which has a 5-HT_2A_ antagonistic effect, reduced the severity of anxious behavior in animals both in conflict conditions (Vogel conflict test) and a non-conflict (Elevated Plus Maze) stressful situation, which is comparable to the effect of diazepam [15]. For 2-mercaptobenzimidazole derivatives under the code AZH-57-67, an anxiolytic action was shown in Elevated Plus Maze and Open Field tests [16]. Derivatives of 2-aminomethylbenzimidazole showed high analgesic and anti-inflammatory activity in the models “Acetic acid-induced writhing” and “Carrageenan-induced paw edema” in mice, comparable to nimesulide at a dose of 50 mg/kg [17,18,19]. Compound RU-1205is a new highly selective central kappa-opioid agonist, for which an analgesic activity has been previously shown [20], with antiepileptoform and anticonvulsant components, confirmed in isolated experiments with maximal electroshock neurons [21]. The anxiolytic potential of benzodiazepine derivatives has been confirmed by many years of clinical practice, and according to the data of modern research, the neuropsychotropic properties of these compounds are used in many areas of medical science. According to the literature, some diazepine derivatives (TR) have analgesic activity by modulating TRPV_1_ receptors [22]. The selective agonist of GABA_A_ receptors, acting mainly on the α_2/3_ subunit, KRM-II-81, exhibits not only anxiolytic and anticonvulsant effects in standard methods, but also analgesic activity in the formalin test [23].

In previous studies, the anxiolytic potential of diazepino[1,2-*a*]benzimidazole derivatives has been shown [24]. As a product of a combination of privileged benzimidazole and diazepine scaffolds, these derivatives have an anti-anxiety effect that is superior to that of diazepam at an equimolar dose, and less of the pronounced side effects of classical benzodiazepines, such as muscle relaxation, addiction, and daytime sleepiness [25,26]. For example, an anti-anxiety activity has been shown for a number of diazepino[1,2-*a*]benzimidazoles [24], including 11-*p*-fluorophenacyl- and 11-pyrrolidinoethyldiazepinobenzimidazoles. It was found that compound DAB-19 exhibits pronounced anxiolytic activity in the Elevated Plus Maze test [25], in addition to an anticonvulsant effect in the “Pentylentetrazole-induced seizures” model [27]. In addition, compound DAB-31 has analgesic and anxiolytic activities (Table 1) [28,29].

Thus, it was of interest to us to conduct a primary screening study for the anti-anxiety and analgesic activity of 11-substituted-2,3,4,5-tetrahydro[1,3]diazepino[1,2-*a*]benzimidazole (**2a**–**d**) and already known *peri*-condensed analogs (**3a**–**e**) of this tricyclic system (Figure 1, Table 2) [30].

## 2. Results and Discussion

### 2.1. Chemistry

Compounds **2a-d** were obtained by alkylation of 2,3,4,5-tetrahydro[1,3]diazepino[1,2-*a*]benzimidazole **(1a**) and its8,9-dimethylsubstituted analogue (**1b**) with 4-chlorobenzyl bromide (for **2a**), 4-chloroacetic acid fluoroanilide (for **2b,c**) and 4-*tert*-butylphenacyl bromide (for **2d**) in a neutral medium. Compounds **3a-e** were synthesized earlier by cyclization of 11-phenacylsubstituted diazepinobenzimidazoles upon heating in conc. HBr.

### 2.2. Pharmacology

#### 2.2.1. Elevated Plus Maze Test

In the EPM test, in terms of the time spent in the open arms, the effect of all compounds under codes **2** and **3** significantly exceeded that for the control group; however, statistically significant differences were not observed in comparison with diazepam (Figure 1). Furthermore, after the introduction of ketone **2d**, the number of animal entries into the open arm did not increase. The reactivity time of mice, which characterizes the animal’s initial decision rate and the degree of its inhibition, was statistically significantly reduced relative to the control under the action of compounds **2b**, **c**, and **3d**. For the entire line of substances, the decision time was shortened compared to diazepam. The total number of transitions was also increased for the entire series of compounds, both in relation to the control and to the diazepam, which may indicate a certain stimulating effect of the substances on the animals’ locomotor activity. The number of overhangs, as an additional anxiolytic parameter, was significantly higher for all diazepino[1,2-*a*]benzimidazoles **2** and **3** compared to controls, and for most relative to the comparison drug. The number of boluses, as the indicator of the animals’ emotional state, was decreased and was at the level of diazepam, which may indicate an animal’s calmer behavior.

#### 2.2.2. Open Field and Spontaneous Locomotor Activity Assessment

In the OF test, the locomotor activity of mice was higher under the influence of all compounds, except **3b**, compared with controls (Figure 2). Under the influence of diazepam, due to its muscle relaxant effect, the locomotor activity was generally less intense than that in the group of control animals. The number of times the animals reared under the influence of **3d** remained at the level of the control values. The use of compounds **2c**, **d**, and **3** led to an increase in the search activity compared with the control group, which does not exclude their influence on the cognitive functions of experimental animals. In general, there was a slight decrease in the number of short self-grooming acts associated with anxiety, and an increase in the number of long-term self-grooming acts, reflecting the comfortable, calm behavior of the animal. The number of entries to the illuminated center of the OF test remained at the level of control values under the action of compounds **2a**, **3a**, and **3e**. The muscle relaxation effect of compounds **2** and **3** was not indicated in Rotarod, Grid, and Wire tests in comparison with the controls (Figure 3), except for chlorobenzyl derivative **2a**, which significantly reduced the mice retention time on the rotating rod relative to the control group in the Rotarod test (*p* ≤ 0.05).

#### 2.2.3. Tail-Flick Test

According to the results of the TF test for diazepino[1,2-*a*]benzimidazole derivatives **2** and **3**, the analgesic activity of varying severity was shown. No analgesic effect was shown for substances **2a**, **c** and **3b**. Thus, after the introduction of these compounds, the observed indicator remained at the level of control values (4.2 ± 0.09 s), and the value of the MPE did not exceed 10%. In the groups of animals treated with compounds **2b**, **d**, **3a**, **c**–**e**, there was a statistically significant increase in the latent period of tail flicker compared with the control group, which suggests the presence of an analgesic effect for the studied compounds; however, statistically significant differences from the butorphanol group were not observed. In addition, the maximum possible effect from the administration of compounds **3d**, **e** was 1.8 times higher than the MPE of the reference drug butorphanol tartrate (*p* ≤ 0.05). Data on the analgesic activity of the studied compounds in the TF test are presented in Figure 4.

#### 2.2.4. Hot Plate Test

According to the results of the HP test, an increase in the latent period of licking the hind paw was noted for all experimental groups. The indices of compounds **2a**, **3a**, **d** were inferior to the group of the comparison drug; in groups **2b**, **3b**, **c**, **e** the values of the latent period of licking of the hind paw were determined at the level of butorphanol. Significant differences from the control values and superiority over the comparison drug group were established in groups **2c**, **d**. Data on the analgesic activity of the compounds under study in the HP test are presented in Figure 4.

#### 2.2.5. Structure–Activity Relationships of Compounds 2 and 3

According to the results of the screening study for diazepino[1,2-*a*]benzimidazole derivatives **2** and **3**, it was shown that these derivatives have anxiolytic and analgesic effects of varying severity. Among 11-substituted diazepino[1,2-*a*]benzimidazoles**2**, ketone **2d** and amide **2b** exhibited the highest analgesic activity. Amide **2c**, a close analogue of compound **2b**, containing methyl groups at positions 8 and 9, and chlorobenzyl derivative **2a**, did not show a pronounced analgesic effect. Based on the data obtained, it is possible to construct a gradation of the analgesic effect: **2d**>**2b**>**2c**>**2a**, from which it follows that the presence of a phenacyl fragment substituted in the phenyl nucleus by a non-polar *tert*-butyl radical in **2d** significantly enhances the analgesic properties, and upon substitution in this position for Cl or F atoms, the activity decreased. The introduction of methyl groups at positions 8 and 9 (**2c**) did not affect the level of analgesic activity. In the HP test, the MPE (%) for compound **2** was calculated at the level of, or even higher than, the reference drug butorphanol, which may indicate the predominant effect of compounds **2** on the supraspinal mechanisms of pain regulation.

Furthermore, the introduction of substances **2a**–**d** led to the development of an anxiolytic effect, which statistically significantly exceeded the indicators of the control group in terms of the time spent in the open arms of the EPM; however, it did not reach the level of diazepam. By the parameter of the number of entries into open arms of the EPM for compounds **2a**–**c**, the level of control values was statistically significantly exceeded, also exceeding the indices of diazepam for **2a**, **c**. In terms of the number of overhangs, all compounds of this series showed an effect that was not inferior than this for diazepam, and had a psychostimulating effect on mice in terms of locomotor activity and the number of times the mice reared in the OF compared to the control group. Thus, the presence of Cl and F radicals in the phenyl nucleus of the phenacyl fragment of structures **2a** and **2c** had a positive effect on the anti-anxiety effect of the substances, in contrast to the pattern previously identified for analgesic properties.

Among the *peri*-cyclic structures **3**, the highest analgesic effect was observed when compound **3e** was administered to mice, which contained a fluorine atom in the aryl substituent and methyl groups in the phenyl nucleus. The replacement of the fluorine atom with the hydroxyl group in the structure of substance **3a** and the methyl groups for hydrogen atoms led to a decrease in the analgesic effect compared to **3e**. Nevertheless, in the TF test, **3a** exhibited an activity that was statistically significantly higher than the control values and slightly higher than that of butorphanol, in addition to **3c**, which, like **3a**, contains hydrogen atoms in the phenyl nucleus, but a biphenyl substituent at position 4 instead of hydroxyphenyl. Compound **3b**, 4-fluorosubstituted at the aryl radical, in contrast to hydroxysubstituted **3a**, did not show an analgesic effect in the TF test. The structural analogue of the most active compound **3e** under the code **3d**, which differs by a hydroxyl substituent instead of the F atom of the aryl radical, in turn, did not show pronounced analgesic activity in the HP test. According to the results of calculating the MPE (%), in the TF test in compounds **3**, all compounds, excluding **3b**, exhibited an effect at the level of butorphanol, which may indicate the effect of compounds **3** on the spinal mechanisms of nociception.

Compounds **3a**–**d** showed a statistically significant difference from the control group in terms of time and number of entries into the open arms of the EPM, in addition to the number of overhangs. At the same time, only the level of anti-anxiety action of fluorophenyl derivative **3b**, which did not show an analgesic effect in TF and HP tests, corresponded to the reference drug diazepam. For compound **3b**, there was also no statistically significant increase in the motor activity in the OF compared to the control, and the number of entries to the center of the OF was doubled relative to diazepam, which is a positive criterion for positioning **3b** as a new anxiolytic. For oxyphenyl derivative **3a**, a close structural analogue of **3b** with moderate analgesic properties, the time spent by the animals in the open arms of the EPM installation decreased 2–3 times compared with **3b**. The anti-anxiety effect of substance **3e**, which showed the most pronounced analgesic properties in compounds **3**, was found to be low, and in terms of the time spent by animals in open arms, did not exceed the control values.

#### 2.2.6. In Silico Analysis

In order to detail the targeted nature of the anxiolytic effect of the most active compound **3b**, an in silico analysis of the molecular binding mechanism to the GABA-binding and benzodiazepine sites of the GABA_A_ receptor, as well as to the specific and allosteric sites of the 5-HT_2A_ receptor, was carried out in comparison with diazepam and ketanserin reference drugs, respectively.

The calculated values of the docking energies and binding constants of the studied compounds are shown in Table 3.

According to the data in Table 3, compound **3b** preferentially interacts with the benzodiazepine site of the GABA_A_ receptor, with a lower affinity for its GABA binding site.

Compared to **3b**, diazepam, in terms of binding constants, interacts 5.4 times weaker with the benzodiazepine site of the GABA_A_ receptor and 2 times weaker with its GABA binding site.

In addition, in accordance with the obtained ΔE and K values, compound **3b** interacts very intensively with the 5-HT_2A_ receptor; in terms of its affinity, **3b** is practically comparable to ketanserin.

At the same time, diazepam exhibits a significantly lower affinity for the 5-HT_2A_ receptor: in comparison with ketanserin, in terms of binding constant values, it interacts 10.5 times weaker with the specific site of 5-HT_2A_ and 20.5 times weaker with its allosteric site.

Thus, according to in silico data, compound **3b** should have a higher anxiolytic activity than diazepam, which is due to the significantly higher affinity of **3b**, in comparison with diazepam, to both the benzodiazepine and GABA-binding sites of the GABA_A_ receptor, as well as a very pronounced 5-HT_2A_ antagonistic action comparable to that of ketanserin and significantly greater than the serotonergic action of diazepam.

The comparison of the binding pattern with the benzodiazepine and GABA-binding sites of the GABA_A_ receptor for compound **3b** and the reference drug diazepam is shown in Figure 5.

Comparison of the binding sites reveals that **3b** and diazepam have very similar mechanisms of binding to the benzodiazepine site: five of the six amino acids involved in interactions are common: PHE757D, TYR817D, TYR867D, TYR1477E, and PHE1496E. TYR1477E participates both in the electrostatic interaction with nitrogen of the diazepine ring and in the hydrophobic interaction with the fluorophenyl fragment in the case of **3b**, and in the case of diazepam, only in the hydrophobic interaction with the Cl substituent. PHE1496E is involved only in hydrophobic interactions in the case of **3b**, and in the case of diazepam, both hydrophobic interactions and pi-stacking. MET1549E is involved in hydrophobic binding in **3b**, which is not characteristic of diazepam.

In the case of the GABA-binding site, both **3b** and diazepam molecules exhibit numerous hydrophobic interactions. The similar binding mechanism of **3b** and diazepam is confirmed by the presence of four common binding amino acids: LEU785D, THR829D, THR783D, and ALA1871K. THR779D and TYR1850K are involved in hydrophobic binding in **3b**, while in diazepam, it is only THR1199C.

The higher affinity of **3b** compared to diazepam is probably due to the greater number of hydrophobic interactions and strong electrostatic interactions and, as a result, higher total energy.

Consideration of the spatial arrangement of these two compounds at the benzodiazepine site of the GABA_A_ receptor shows that **3b** is characterized by four-center binding, while only three interaction centers are involved in diazepam fixation. These features provide a stronger fixation of the **3b** molecule in the site, which leads to a higher activity of this compound. At the GABA-binding site, the spatial arrangement of **3b** and diazepam is approximately the same and for both molecules, is due to three-center hydrophobic binding (Figure 6).

Comparison of the binding pattern with the specific and allosteric sites of the 5-HT_2A_ receptor for compound **3b** and the reference ketanserin is shown in Figure 7.

The ketanserin and **3b** molecules include fluorinated phenyl, which provides hydrophobic interactions with the same identical seven amino acids at a specific 5-HT_2A_ site. Thus, in this area of interaction, the binding mechanisms of **3b** and ketanserin are the same.

The second benzene ring in the ketanserin structure is stacked and provides hydrophobic interactions, which seem to provide some greater affinity of ketanserin for the specific site 5-HT_2A_, compared to **3b**.

In the case of the allosteric site of the 5-HT_2A_ receptor, the binding mechanisms of **3b** and ketanserin are practically the same: both molecules have two broad regions of hydrophobic interactions by means of six common binding amino acids.

The spatial arrangement of these two compounds at the specific site of the 5-HT_2A_ receptor is approximately the same and confirms the assumptions made: **3b** has three regions of hydrophobic interaction, while ketanserin has three regions of hydrophobic interaction and pi-stacking. However, in both cases, three-center binding is observed. In the allosteric site of 5-HT_2A_, the spatial arrangement of **3b** and ketanserin is also approximately the same and for both molecules, is due to three-center hydrophobic binding (Figure 8).

The results of the performed in silico analysis with a high degree of probability allow us to assert that the high anxiolytic activity of compound **3b** is explained by the combination of a pronounced interaction mainly with the benzodiazepine site of the GABA_A_ receptor with a prominent interaction with both the specific and allosteric sites of the 5-HT_2A_ receptor.

## 3. Materials and Methods

### 3.1. General Procedure for the Synthesis of Compounds 2

Compounds **2a**–**d** were obtained by alkylation mainly in the 1*H*-tautomeric form of 2,3,4,5-tetrahydro[1,3]diazepino[1,2-*a*]benzimidazole (**1a**) [31] or its 8,9-dimethylsubstituted analog (**1b**) 4-chlorobenzyl bromide, 4-chloroacetic acid fluoroanilide, and 4-*tert*-butylphenacyl bromide in neutral medium. Compounds **3a**–**e** were cyclized and synthesized earlier [30] with 11-phenacylsubstituted diazepino[1,2-*a*]benzimidazoles upon heating in conc. HBr.

#### 3.1.1. 11-(4-chlorobenzyl)-2,3,4,5-tetrahydro[1,3]diazepino[1,2-*a*]benzimidazole Hydrochloride (**2a**)

A solution of 1.87 g (10 mmol) of 2,3,4,5-tetrahydro[1,3]diazepino[1,2-*a*]benzimidazole (**1a**) and 1.60 g (10 mmol) of 4-chlorobenzyl chloride in 10 mL of nitromethane was boiled for 5 h. After cooling, the separated precipitate was filtered, washed with acetone. Yield 3.12 g (90%). Colorless crystals, m.p. 231–232 °C (was recrystallized from EtOH).^1^H NMR spectrum (300 MHz), δ, ppm (DMSO-*d*_6_): 1.96–2.10 (m, 4H, C(3)H_2_, C(4)H_2_), 3.56–3.58 (m, 2H, C(2)H_2_), 4.32–4.36 (m, 2H, C(5)H_2_), 5.47 (s, 2H, CH_2_Ar), 7.30–7.42 (m, 7H, H(8), H(9), H(10) or H(7), H(2′), H(3′), H(5′), H(6′)) (Hereinafter, numbers with a stroke show the protons of the aryl substituent in position 11 of diazepinobenzimidazole **1**), 7.71 (d, 1H, H(7) or H(10), *J* = 7.5), 8.99 (s, 1H, ^+^NH). Found, (%): C 62.34; H 5.21; Cl 20.00; N 12.36. C_18_H_18_ClN_3_^.^Cl. Calculated, (%): C 62.08; H 5.50; Cl 20.36; N 12.07 (Figure 9).

#### 3.1.2. N-(4-fluorophenyl)-2-(2,3,4,5-tetrahydro[1,3]diazepino[1,2-*a*]benzimidazol-11-yl)acetamide Hydrochloride (**2b**)

A solution of 0.38 g (2 mmol) of diazepinobenzimidazole **1a** and 0.38 g (2 mmol) of 4-fluoroanilide of chloroacetic acid in 5 mL of nitromethane was boiled for 6 h, cooled, 5 mL of diethyl ether was added, and the precipitate that formed was filtered off and washed with acetone. Yield 0.57 g (75%). Colorless crystals, m.p. 231–232 °C (was recrystallized from CH_3_CN). ^1^H NMR spectrum, (600 MHz), δ, ppm (DMSO-*d_6_*): 1.98–2.09 [m, 4H, C(3)H_2_, C(4)H_2_], 3.52–3.46 [m, 2H, C(5)H_2_], 4.34–4.35 [m, 2H, C(2)H_2_], 5.22 (s, 2H, CH_2_CO), 7.16 [t, 2H, H(8), H(9), *J* = 8.7 Hz], 7.33–739 [m, 2H, H(2′), H(6′)], 7.60 (d, 1H, H(10), *J* = 8.0 Hz), 7.62–7.65 [m, 2H, H(3′), H(5′)], 7.69 [d, 1H, H(7), *J* = 7.8 Hz], 9.62 (s, 1H, NHCO), 10.96 (s, 1H, ^+^NH). Found, (%): N 19.68. C_19_H_19_FN_4_O^.^HCl. Calculated, (%): N 19.95 (Figure 10).

#### 3.1.3. N-(4-fluorophenyl)-2-(8,9-dimethyl-2,3,4,5-tetrahydro[1,3]diazepino[1,2-*a*]benzimidazol-11-yl)acetamide Hydrochloride (**2c**)

This compound was prepared analogously to compound **2b** from 8,9-dimethyldiazepinobenzimidazole **1b** and chloroacetic acid 4-fluoroanilide in 79% yield. Colorless crystals, m.p. 260–262°C (was recrystallized from propanol-2). ^1^H NMR spectrum, (600 MHz), δ, ppm (DMSO-*d*_6_): 1.95–2.04 [m, 4H, C(3)H_2_, C(4)H_2_], 3.29 (s, 3H, CH_3_), 2.31 (s, 3H, CH_3_), 3.47–3.52 [m, 2H, C(5)H_2_], 4.25–4.27 [m, 2H, C(2)H_2_], 5.18 (s, 2H, CH_2_CO), 7.14–7.17 [m, 2H, H(2′), H(6′)], 7.40 [s, 1H, H(10)], 7.49 [s, 1H, H(7)], 7.63–7.65 [m, 2H, H(3′), H(5′)], 9.12 (s, 1H, NHCO), 10.97 (s, 1H, ^+^NH). Found, (%): N 13.67. C_21_H_23_FN_4_O^.^HCl. Calculated, (%): N 13.91 (Figure 11).

#### 3.1.4. 11. -(4-tert-butylphenyl)-2-(2,3,4,5-tetrahydro[1,3]diazepino[1,2-*a*]benzimidazol-11-yl)ethanone Hydrobromide (**2d**)

A solution of 1.87 g (10 mmol) of 2,3,4,5-tetrahydro[1,3] diazepino[1,2-*a*]benzimidazole **1a** and 2.55 g (10 mmol) of 4-*tert*-butylphenacyl bromide in 10 mL of acetonitrile was boiled for 10 h. After cooling, the separated precipitate was filtered off and washed with acetone. Yield 3.75 g (85%). Colorless crystals with m.p. 217–220 °C (was recrystallized from CH_3_CN).^1^H NMR spectrum, (600 MHz), δ, ppm (DMSO-*d_6_*): 1.31 [s, 9H, C(CH_3_)_3_], 1.92–1.95 [m, 2H, C(3)H_2_], 2.05–2.08 [m, 2H, C(4)H_2_], 3.44–3.46 [m, 2H, C(5)H_2_], 4.34–4.35 [m, 2H, C(2)H_2_], 5.94 (s, 2H, CH_2_COAr), 7.32 (t, 1H, H(8) or H(9), *J* = 7.4 Hz), 7.35 (t, 1H, H(9) or H(8), *J* = 7.8 Hz), 7.59 (d, 1H, H(10), *J* = 8.0 Hz), 7.63 (d, 2H, H(3′), H(5′), *J* = 8.5 Hz), 7.70 (d, 1H, H(7), *J* = 7.8 Hz), 7.99 (d, 2H, H(2′), H(6′), *J* = 8.5 Hz), 8.77 (s, 1H, ^+^NH). Found, (%): C 62.02; H 6.04; Br 18.31; N 9.23. C_23_H_27_BrN_3_O^.^HBr. Calculated, (%): C 62.44; H 6.38; Br 18.06; N 9.50 (Figure 12).

## 4. Experimental Procedure

### 4.1. Animals

The experiments were carried out on 330 adult male BALB/c mice, with an average weight of 18–25 g. Mice were randomly divided into equal groups of 6 animals. The animals were kept in a vivarium with a 12/12 light/dark cycle on a standard food diet for laboratory animals without restriction of access to food and water, in compliance with the International Recommendations for the Protection of Vertebrates Used in Experimental Research (2017). The performance of the work corresponds to the parameters for preclinical studies (Order of the Ministry of Health of Russia dated 04.01.2016 N 199n “On the approval of the Rules of Good Laboratory Practice”, registered with the Ministry of Justice of Russia on 08.15.2016 N 43232).

### 4.2. Drugs and Treatment

Diazepino[1,2-*a*]benzimidazoles **2** and **3** were synthesized by the Research Institute of Physical and Organic Chemistry, Southern Federal University, Rostov-on-Don, Russia. The control groups were injected with a solvent (distilled water) in an equivalent volume of 0.1 mL per 10 g of an individual’s weight. The animals of the reference drug groups received diazepam (J.S.C. Polfa, Poland) at a dose of 2 mg/kg [29] and butorphanol (FSUE “Moscow Endocrine Plant”, Russia) at a dose of 1 mg/kg [20]. Substances of groups **2** and **3** were injected per oral with an atraumatic metal probe 30 min before the start of the experiments. Their doses were calculated equimolar to the comparison drug diazepam (2 mg/kg) in the Elevated Plus Maze, Open Field, Rotarod, Grid and Wire tests. For the Tail Flick and Hot Plate tests, compounds were administered i.p. to mice 60 min before the start of the experiments, and doses **2** and **3** were calculated equimolar to the reference drug butorphanol (1 mg/kg).

### 4.3. Anxiolytic Assay

The anxiolytic activity of the compounds was studied in the Elevated Plus Maze test [32]. The mouse was placed in the center of the installation with its head towards the open arm; the observation time for each animal was 5 min. The following parameters were recorded: reactivity—the time of the first exit of the animal from the center of the installation (s), the number of transitions between the arms, the time spent in open arms (s), the number of entries into open arms, the number of overhangs, and the number of fecal boluses.

The study of animal’s behavior was carried out in an Open Field test [33]. During 5 min of the test, the locomotor activity (the number of crossed quadrants) and the number of times the animal reared, in addition to the search activity (represented by the number of times the animal peeped into the holes), the animals’ reactivity time (the time of leaving the central segment, s), the number of entries to the center of arena, and the number of acts of short (less than 5 s) and long (more than 5 s) self-grooming were recorded.

### 4.4. Analgesic Assay

The search for analgesic-active compounds was carried out in Tail Flick and Hot Plate tests (UgoBasile, Varese, Italy) [34]. In the first test, the tail withdrawal time (s) from thermal stimulation (55 °C) was estimated; in the second test, the animal was placed on a plate heated to 55 °C surrounded by a transparent cylinder, and then the latent time of the first licking of the hind paw was recorded (s). To avoid the risk of damage to animal tissue, the maximum duration of thermal exposure was limited to 15 s in the Tail Flick test and 30 s in the Hot Plate test.

For each group of studied substances, the value of the maximum possible effect (*MPE*, %) was determined, calculated by the formula [35,36]:*MPE* (%) = (*LP*_2_ − *LP*_1_)/(*MAX_TIME_*− *LP*_1_) ∗ 100% (1) where *LP*_2_is the latent period of the reaction after the administration of the substance, *LP_1_* is the latent period of the reaction before the administration of the substance, and *MAX_TIME_* is the maximum time of the stimulus application.

### 4.5. Muscle Relaxant Properties

The study of muscle relaxant effects of substances was carried out in Rotarod, Grid, and Wire tests [29]. The Rotarod device is a rod with a diameter of 2.5 cm rotating at a constant speed of 10 rpm. The mouse was placed on the rod and the retention time on it was recorded—maximum 30 s. For the Grid test, the animals were placed on a horizontal grid from the bottom side (upside down). During 30 s of the test, the number of paws used by the animals for holding was recorded. Similar indicators were recorded for the Wire test, where the animals had to pull themselves up on a wire, which they held with two paws. The Grid and Wire tests were assessed in points from 0 to 4 according to the number of paws involved.

### 4.6. In Silico Analysis

Construction of the optimized 3D-models of compounds was performed using molecular mechanics and quantum chemistry successively with programs MarvinSketch 17.1.23 [37] and MOPAC2016 [38].

Search for valid 3D-models of the most abundant α1-β2-γ2-GABA_A_ subtype receptor and 5-HT_2A_ receptor of *Homo sapiens* was conducted using a previously developed method [39]. For each biotarget, three such models were found: for the GABA_A_ receptor, two experimental 6D6T, 6D6U [40] and one theoretical [41], for the 5-HT_2A_ receptor also two experimental 6A93, 6A94 [40] and one theoretical 5d6b98e9e00432ae919177b2b38d636c [42].

Ensemble docking was performed using the AutoDockVina 1.1.2 program [43], each compound in 10 conformers, 5 times in each site of each model, with the calculation of 150 obtained values of the minimum docking energies ΔE, as described in [39]. The binding mechanism was analyzed using the LigandScout 4.1 program [44].

## 5. Statistical Analysis

Statistical processing of the obtained data *in vivo* was carried out using the Kolmogorov–Smirnov test with the Wilcoxon–Lilliefors test. In the case of normal distribution of the data, one-way ANOVA was used with Dunnett’s post-hoc test; for non-parametric statistics, the Kruskal–Wallis test and Dunn’s post-hoc test were used. Calculations were implemented in GraphPad Prism v.7.0. Results are presented as M ± SEM (*p* ≤0.05).

## 6. Conclusions


Derivatives of diazepino[1,2-*a*]benzimidazole **2** and **3** showed anxiolytic and analgesic activities of varying severity.The presence of the 2,3,4,5-tetrahydro[1,3]diazepino[1,2-*a*]benzimidazole fragment in the structure of the substances tends to manifest a moderate anxiolytic effect (**2a**–**d**) in terms of the time spent in the open arms of the EPM. According to the totality of anxiolytic parameters in the series **2a**–**d**, chlorobenzyl derivative **2a** and fluorophenylacetamide **2c** were the most active. In Open Field, **2a**–**d** had a psychostimulating effect. The presence of electronegative Cl and F atoms in the radical part of the structure of the compounds was found to be less favorable for analgesic activity than for anti-anxiety activity.In compounds **3**, according to the parameters characterizing the anxiety state of animals (the number of entries and time spent in open arms, the number of overhangs), the substances under the code **3a**–**e** showed a statistically significant difference from the control group; however, the effect at the level of diazepam was shown only for the fluorophenyl derivative **3b**. Administration of compounds **3a**–**e** positively influenced the search activity of mice in Open Field.For the analgesic action of compounds of series **2**, the most effective was the substitution of (4-fluorophenyl)acetamide (**2b**) and 4-*tert*-butylphenacyl (**2d**), in addition to the hydrogen atoms in positions 8 and 9. Derivatives of **2** mainly affect the supraspinal regulation of pain, and **3** mainly affects the spinal regulation.In compounds **3**, the most significant aspect for the analgesic effects was the presence in the molecule of methyl substituents at positions 8 and 9, and a 4-fluorophenyl substituent of the core structure (**3e**). Compounds **3a** and **3c**, containing hydrogen atoms instead of methyl substituents (**3e**) and an aryl or hydroxyl substituent in the *p*-position of the phenyl radical, had a somewhat less pronounced analgesic effect.According to the in silico study, **3b** is characterized by the interaction with the GABA_A_ receptor and by a prominent 5-HT_2A_ antagonistic action.

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
