# Peer review of "Synthesis and Pharmacological Evaluation of Novel 2,3,4,5-tetrahydro[1,3]diazepino[1,2-a]benzimidazole Derivatives as Promising Anxiolytic and Analgesic Agents"

_molecules, 2021, doi:10.3390/molecules26196049_

Round 1

Reviewer 1 Report

This is a classic paper in medicinal chemistry presenting physiologic activity of two series of novel compounds - in this case on anxiolytic and analgesic properies. Thus, one set of compouds was syntesized for the need of this paper, whereas second set was available from Authors laboratories. Their physiologic activity was tested by the appropriate standard methods and some computer-aided study was done in order to clearify their possible mode of action. Thus, paper is suitably planned and falls into Journal scope. However, it is not well written and have to be significantly reconstructed before publication. Below are some comments, which should help in reorganization of the paper:

1./ Authors use full chemical names to describe each of the studied compounds. I would like to propose to use numbering and appropriate figures. Full names are difficult to follow and made the text difficult to read;

2./ Formulas of all RUs discussed in the text should be given. What is the relation of RU-31 with others? Was it the first one of such activity or is already commercialized? This part of the manuscript, which in fact presents rationale of studies, should be changed.
I would propose to add a Figure with structures of all the compounds described in this part of text and number each of the compound in order to avoid the use of long chemical names as sole determinants of active molecules;

3./ Figure presenting synthetic scheme should be moved from Supporting Inforlmation to main body of the manuscript. Additionally it should be corrected: (i) add a formula of tautomeric form of compounds 1, (ii) there is no need to differentiate between compounds 1a and 1b. General formula of both is the same. The same considers paragraph 3.1.;

4./ Figures S1 and S2 (by the way there shoud be activity not sctivity in one of plots) and table T1 should be moved to main body of the manuscript.

5./ remove "(M ± SEM)". from text in page 4;

6./ introduce part of Figure S4 devoted to hot plate test into main body of the manuscript;

7./ remove formula and comment considering binding contsat K from the text - this is simply obvious;

8./ I propose to rewrite the paragraph devoted to molecular modeling by  resigning from detailed description of binding of 3b and reference drugs to the subsequent binding sites and concentrate on comparisons (comparisions are described well). This will shorten the text and it will be cleared. Also move  Figures S6 and S8 to the main body of the manuscript.

9./ Authors please rewrite paragraph 3.2. because its written really badly;

10./ does m.p. 231–232 °C (EtOH) means that the compound was crystallized from ethanol? If so it should be specified;

11./ When numbering carbon atoms of heterocyclic system for description of NMR spectra, appropriate scheme of numbering should be given (maybe in Figure presenting synthetic scheme?);

12./ in the manuscript body diazepan is wrtitten as Diazepam, in the Experimental as diazepam - please normalize;

13./ Homo sapiens not Homo sapiens (paragraph 4.6.);

14./ in Conclusions give full name of OF test;

15./ using forms "row 1" and "row 2" when presenting sets of compounds is somewhat unusual.

Author Response

Dear Reviewer,

We are very grateful to you for the work done.

Authors propose to change the name of article to Synthesis and Pharmacological Evaluation of Novel 2,3,4,5-tetrahydro[1,3]diazepino[1,2-a]benzimidazole Derivativesas Promising Anxiolytic and Analgesic Agents. Would it be better?

As for RU-31, we would like to clarify. Being derivatives of imidazobenzimidazole, compounds of the RU series are examples of benzimidazoles with analgesic and anxiolytic activities. Hereinafter, examples of diazepines with similar properties are also given. Thus, we bring the reader to the thesis that the combined scaffolds have the sought activity and the hypothesis that the combination product has similar properties, possibly with mitigated undesirable reactions. A table with the structures of the studied substances has been added to the introduction as you proposed.

The authors sincerely thank the reviewer for the comments made, describing the results in silico. The simple description of the linking details has been removed from the text of the article, only a comparative description of the interaction is left. Part of the text has been corrected in this regard. Indeed, reducing the description and focusing on comparison significantly increased the substantive understanding of the results obtained. 

We also decided to move all schemes, tables and figures to the main text. All the changes on chemical part of the study were made accorded to your revisions.

Authors have a question. In paragraph 9 you advised to rewrire paragraph 3.2, but our work doesnt content this number. Could you please clarify?

Thank you!

Reviewer 2 Report

The manuscript titled 'Study of Anxiolytic and Analgesic Properties...' by D. V. Maltsev at al. has as main subject the study and synthesis of some diazepines as novel possible medicines. The manuscript contains a large amount of data and it is completed by a Supplementary material, also containing a large number of Figures, used to support the main text. The work is well documented and conducted in a scientific way. The results and conclusions are supported by experimental data, Details of the experiments are enough to make them reproducible. In the experimental synthetic details, sometimes melting point is abbreviated as m.p and sometimes as m.t, please correct. In the Supplementary materials, please reduce the size of Schemes and allow a table not to be split between two pages. The main issue with this work is the absence of any Figure or Scheme inside the manuscript, all of them are presented as Supplementary materials. This make the work unattractive and difficult to handle, because the readers has to switch very often between the two documents. Therefore, the work should be improved by inserting the important figures form the supplementary material inside the main text. After these corrections the paper will become appropriate for pblication.

Author Response

Dear Rewiewer,
We are very grateful to you for the work done.
All your suggested edits have been accepted. Schemes, tables and figures also have been moved to the main text.
Thank you!

Round 2

Reviewer 1 Report

Now paper looks good and could be published as it is. There are space between words lacking in lines 83, 103,494,,498 and 503, but these errors could be corrected upon final proofing of the manuscript.